Ozymandias: a biodiversity knowledge graph

Page Roderic D.M. roderic.page@glasgow.ac.uk rdmpage@gmail.com
IBAHCM, MVLS, University of Glasgow , Glasgow , United Kingdom
Lapp Hilmar
Electronic publication date: 2019 Apr 8
Publication date: 2019
Volume: 7
Electronic Location ID: e6739
Received 2018 Dec 19; Accepted 2019 Mar 7
Copyright: ©2019 Page
Copyright year: 2019
Copyright holder: Page
License: This is an open access article distributed under the terms of the Creative Commons Attribution License, which permits unrestricted use, distribution, reproduction and adaptation in any medium and for any purpose provided that it is properly attributed. For attribution, the original author(s), title, publication source (PeerJ) and either DOI or URL of the article must be cited.
License URL: https://creativecommons.org/licenses/by/4.0/

Keywords: Knowledge graph, Biodiversity informatics, Linked data, Identifiers, Challenge

Funding: The authors received no funding for this work.

==============================
Enormous quantities of biodiversity data are being made available online, but much of this data remains isolated in silos. One approach to breaking these silos is to map local, often database-specific identifiers to shared global identifiers. This mapping can then be used to construct a knowledge graph, where entities such as taxa, publications, people, places, specimens, sequences, and institutions are all part of a single, shared knowledge space. Motivated by the 2018 GBIF Ebbe Nielsen Challenge I explore the feasibility of constructing a “biodiversity knowledge graph” for the Australian fauna. The data cleaning and reconciliation steps involved in constructing the knowledge graph are described in detail. Examples are given of its application to understanding changes in patterns of taxonomic publication over time. A web interface to the knowledge graph (called “Ozymandias”) is available at https://ozymandias-demo.herokuapp.com.

Introduction

“Linnaeus would have been a ‘techie”’ (Godfray, 2007)

The recent announcement that the Global Biodiversity Information Facility (GBIF, https://www.gbif.org) has reached the milestone of one billion occurrence records reflects the considerable success the biodiversity community has had in mobilising data. Much of this success comes from standardising on a simple column-based data format (Darwin Core) (Wieczorek et al., 2012) and indexing that data using three fields: taxonomic name, geographic location, and date (i.e., “what”, “where”, and “when”). By flattening the data into a single table, Darwin Core makes data easy to enter and view, but at the cost of potentially obscuring relationships between entities, relationships that may be better represented using a network. In this paper, I explore the representation of biodiversity data using a network or “knowledge graph”.

A knowledge graph is a network or graph where nodes represent entities or concepts (“things”) and the links or edges of the graph represent relationships between those things (Bollacker et al., 2008). Each node is labelled by a unique identifier, and may have one or more attributes or properties. Each edge of the graph is labelled by the name of the relationship it represents. A common implementation of the basic unit of a knowledge graph is the linked data triple of subject, predicate, and object, where the subject (e.g., a publication) is connected to an object (e.g., a person) by a predicate (e.g., “author”). Triples are not the only way knowledge graphs can be modelled, other options include graph databases and search engines (e.g., Neo4J (https://neo4j.com) and Elasticsearch (https://www.elastic.co)), but adopting triples means we can use existing technologies such as triple stores and the SPARQL query language (W3C SPARQL Working Group, 2013).

Knowledge graphs are potentially global in scope, hence rely on global identifiers. Most datasets will have their own local identifiers for the entities they contain, such as species, publications, specimens, or collectors. These identifiers are adequate for local use, but local identifiers also serve to keep data isolated in distinct silos. Hence we need to map identifiers for the same thing between the different silos. This can be done by establishing a “broker” service that asserts identity between a set of identifiers, or by mapping local identifiers to a single global identifier. The case for mapping to a single global identifier (“strings to things”) is attractive in terms of scalability (mapping each local identifier to a single global identifier is easier than managing cross mappings between multiple identifiers), and is even more attractive if there are useful services built around that global identifier. For example, Digital Object Identifiers (DOIs) are becoming the standard for identifying academic publications. Given a DOI we can retrieve metadata about the work from CrossRef (https://www.crossref.org), we can get measures of attention from services such as Altmetric (https://www.altmetric.com) and we can discover the identities of the work’s authors from ORCID (https://orcid.org). Furthermore, by agreeing on a centralised identifier we effectively decentralise the building of the knowledge graph: given a DOI, anybody that links local information to that DOI is potentially contributing to the construction of the global knowledge graph.

Mapping strings to things give us a way to refer to the nodes in the knowledge graph, but we also need a consistent way to label the edges of the graph. There has been an explosion in vocabularies and ontologies for describing both attributes of entities and their interrelationships. While arguments can be made that domain-specific ontologies enable us to represent knowledge with greater fidelity, the existence of multiple vocabularies comes with the cognitive overhead of having to decide which term from what vocabulary to use. In contrast to, say Senderov et al. (2018), who use several ontologies to model taxonomic publications, the approach I have adopted here is to try and minimise the number of vocabularies employed, and to avoid domain-specific vocabularies where ever possible. For this reason the default vocabulary used is schema.org (https://schema.org), which is being developed by a consortium of search engine vendors including Google, Microsoft, and Yahoo. In addition to simplifying development, adopting a widely used vocabulary increases the potential utility of the knowledge graph. One motivation for the development of schema.org is to encourage the inclusion of structured data in web pages, helping search engines interpret the contents of those pages. By adopting schema.org in knowledge graphs we can make it easier for developers of biodiversity web sites to incorporate structured data from those knowledge graphs directly into their web pages.

There are several different categories of applications that can be built on top of a knowledge graph, for example data reconciliation, data augmentation, and meta-analyses. Reconciliation involves either matching strings to things, or matching entities from different data sources. An example of reconciliation is matching author names to identifiers. Augmentation involves combining data for the same entities from different sources that individually may be incomplete, but together yield more extensive coverage of those entities. An example is supplementing existing imagery of species with figures published in the taxonomic literature. Meta-analyses make use of the data aggregated in the knowledge graph to explore larger patterns. There have been numerous studies of patterns of taxonomic activity (Joppa, Roberts & Pimm, 2011; Costello, Wilson & Houlding, 2013; Bebber et al., 2013; Grieneisen et al., 2014; Mesibov, 2014; Sangster & Luksenburg, 2014; Tancoigne & Ollivier, 2017), typically these studies assembled a custom database, and often this data is not made more widely available, or the data is not actively updated. Having a biodiversity knowledge graph would enable users to ask similar questions but for different taxonomic groups, or different time periods.

In response to the GBIF 2018 Ebbe Nielsen Challenge (https://www.gbif.org/news/1GQURfK5jS4Iq4O06Y0EK4/2018-gbif-ebbe-nielsen-challenge-seeks-open-data-innovations-for-biodiversity) I constructed a knowledge graph for the Australian fauna, based on data in the Atlas of Living Australia (ALA, https://www.ala.org.au) and the Australian Faunal Directory (AFD, https://biodiversity.org.au/afd/home). This regional-scale dataset was chosen to be sufficiently large to be interesting, but without being too distracted by issues of scalability. The knowledge graph combines information on taxa and their names, taxonomic publications, the authors of those publications together with their interrelationships, such as publication, citation, and authorship. Constructing the knowledge graph required extensive data cleaning and cross linking. These steps are described below, and examples of the application of the knowledge graph are discussed. My entry’s name, “Ozymandias”, is a reference to Shelly’s poem of the same name, and is both a play on the nickname for Australia (“Oz”), and an acknowledgement of the hubris involved in attempting to build a comprehensive knowledge graph.

Materials and Methods

Knowledge graph

The general structure of the knowledge graph is based on (Page, 2013; Page, 2016a). The core entities are taxa, taxonomic names, publications, journals, and people. Figure 1 summarises the relationships between those entities.

Figure 1 The knowledge graph model used in Ozymandias.

Nodes in the graph are represented by circles and are labelled with the rdf:type of that node. Nodes are connected by edges in the graph which are labelled by a term from a vocabulary, typically schema.org.

Taxa and taxonomic names were modelled using the TDWG Life Sciences Identifier (LSID) vocabulary based on (Kennedy et al., 2006). Taxa are nodes in a tree representing the taxonomic classification and are instances of the type tc:TaxonConcept. The taxonomic classification is represented by rdfs:subClassOf relationship between parent and child taxa (a child is a rdfs:subClassOf its parent).

Taxonomic names (type tn:TaxonName) are connected to the corresponding taxa using relations from the TAXREF vocabulary (Michel et al., 2017) and are typically either accepted names or synonyms. This vocabulary was adopted because it enables a more direct way of expressing the relationship between taxa and taxonomic names than is possible using the TDWG LSID vocabulary.

Taxonomic names are published in publications, which were represented using terms from the schema.org vocabulary. In cases where the full text of an article is available as a PDF file I make use of the schema:encoding property to link the publication to a schema:MediaObject representing the PDF. Articles are linked to the journals they were published in by the schema:isPartOf property.

Representing ordered lists in RDF is not straightforward, which presents a challenge for expressing relationships such as authorship. Not only is the order of authorship an important feature when formatting a published work for display, it is also useful information when trying to reconcile author names (see below). The approach adopted here is to use the schema:Role type (Holland & Johnson, 2014). Inserting a Role node between two entities enables us to annotate that relationship, for example specifying the time period during which the relationship applies, or the source of the evidence for that relationship (Gawriljuk et al., 2016). Holland & Johnson (2014) note that they expect that most consumers of data modelled using Roles would collapse the Role node to retrieve the direct link between the two entities. To make the collapsing easier the Role node reuses the property used to link to the Role to link to the target entity. Hence, rather than directly connect a publication to an author using, say, the schema:creator property, the creator of a work is a Role, which in turn has the author as its creator property. We can then store the position of author in the list of authors using the schema:roleName property (e.g., “1”, “2”, etc.) (Fig. 2).

Figure 2 An example of modelling order of authorship using schema:Role.

Each author is linked to the article they authored via a schema:Role node, which specifies the order of authorship for each author. In this example, “B Y Main” is the first author, “L W Popple” is the second author.

Identifiers

Identifiers are both central to any attempt to link data together, and at the same time can be one of the major obstacles to creating links. Ideally identifiers should be globally unique, persistent, and each entity would have only a single identifier. In reality, entities may have many identifiers, typically minted by different databases, and identifiers may change, or at least have multiple representations. For example, DOIs may contain upper and lowercase letters, but are actually case insensitive. Some databases may choose to store DOIs in lower case form, others in upper case, or any combination in between. Identifiers typically require dereferencing and the mechanism for this may evolve over time, often for reasons outside the control of the organisation that minted the identifier. DOIs are currently dereferenced (“resolved”) using the web proxy https://doi.org. This proxy recently switched from the HTTP to the HTTPS protocol, so that databases populated before and after this switch may store the same DOI as different strings (the older database using http://, the newer using https://) which complicates matching records in the two databases. To minimise the impact of these kinds of changes, Ozymandias stores identifiers both as URLs (where appropriate) but also as property-value pairs (using schema:PropertyValue) where the schema:value property stores the identifier string stripped of any dereferencing prefix. For example, a DOI (https://doi.org/10.5134/176044) would be stored as a schema:PropertyValue with schema:propertyID “doi” and schema:value “10.5134/176044” (Fig. 3).

Figure 3 Storing identifiers using schema:PropertyValue.

The work has two identifiers, a DOI https://doi.org/10.5134/176044 and a Handle https://hdl.handle.net/2433/176044. These are stored as schema:PropertyValue pairs.

Citations

One paper citing another can be represented by a direct link between two identifiers, for example a link between the DOIs of the citing and the cited work. CrossRef provides lists of literature cited for many of the works in its database, and many of these cited works themselves have DOIs. Hence if we have a DOI for a work we can immediately populate the triple store with citation links. This works well if both works have a DOI, but many taxonomically relevant works do not have these identifiers. Even for those works that do have DOIs, these may not have been available at the time the citing work was deposited by a publisher, for example, if the cited work has only recently been assigned a DOI.

To expand the citation links beyond just those works with DOIs I also generated an identifier for each work modelled on the Serial Item and Contribution Identifier (SICI). This identifier comprised the International Standard Serial Number (ISSN) of the journal, together with the volume, and the starting page. This triple uniquely identifies most articles, and is easy to generate. SICIs were generated for works harvested from the Australian Faunal Directory, and from the lists of literature cited obtained from CrossRef, and were stored as schema:PropertyValue pairs in the same way as DOIs and other identifiers. By matching SICIs it was possible to expand citation links beyond those where both works had DOIs.

Populating the knowledge graph

Perhaps the biggest challenge in constructing a knowledge graph is to map names or descriptions of entities to one or more globally unique identifiers. In some cases the sources of data will already have identifiers. Taxa in the ALA each have a unique identifier (a LSID), as do taxa and publications in the AFD (which use UUIDs). The ALA and AFD share the same taxon identifiers, which makes linking the two databases straightforward. However, these identifiers are local in the sense that they are primary keys for local databases that have been converted into URLs. The knowledge graph can only grow if we use external identifiers that are shared by other databases, or at least map local identifiers onto those external identifiers. For publications this is straightforward in the sense that a publication in a database of Australian animals can be unambiguously mapped onto the publication in, say, a database for Japanese animals. However, a taxon as defined in the Australian Faunal Directory may not correspond exactly to a taxon with the same name in another.

Reconciling works

For the works in AFD, I searched for DOIs using the API provided by CrossRef. If a reference was found the associated DOI was assigned to that reference. CrossRef is not the only registration agency for DOIs, there are several others that are used by digital libraries and publishers, such as DataCite, the multilingual European Registration Agency (mEDRA), and Airiti (華藝數位 ). Most of these agencies lack the discovery services provided by CrossRef, so for these DOIs I harvested the article metadata using a combination of web services and screen scraping, created a local MySQL database to store the metadata, and used that database to match references to DOIs. This database was also used to match articles to other classes of identifiers, such as Handles and URLs.

Australian natural history institutions are significant publishers of biodiversity literature, and much of this has been scanned by the Biodiversity Heritage Library in Australia. As a consequence many of the articles in the knowledge graph were available in my BioStor project (Page, 2011). Identifiers for these articles were found by matching using the BioStor OpenURL service.

Reconciling authors

Multiple approaches were used to match author names to external identifiers. Metadata for DOIs from CrossRef would sometimes include ORCID ids for authors. The ORCID record for each ORCID id was retrieved using the ORCID API and converted to a set of RDF triples linking the identifiers for a work (e.g., DOI) to a person’s ORCID. These triples modelled the order of authorship using schema:Role as described above. Similarly, I parsed Wikispecies pages (https://species.wikimedia.org) and extracted bibliographic records for works identified by a DOI, and constructed triples linking the work to its authors where those authors had their own Wikispecies page. Hence to match authors in the knowledge graph to authors in ORCID or Wikispecies, we can ask whether the same pairing of work and author name appears in both databases. For example, we can retrieve the second author of a work in the knowledge graph and in ORCID by querying by DOI for the work and restricting the value of schema:roleName to “2” (Fig. 4). As a final check we can compare the author names and accept only those names whose similarity exceeds a threshold. In this way we can automate the matching authors across databases.

Figure 4 Matching author records in two different databases.

In this example the article with DOI 10.11646/zootaxa.4001.1.1 occurs in both Ozymandias (OZ) and ORCID. Using a SPARQL query we retrieve the name of the second author in the two databases: “L W Popple” in Ozymandias and “Lindsay W Popple” in ORCID. Given the similarity in the names, we conclude that the two authors are the same, and we can assign the ORCID for Lindsay W Popple (https://orcid.org/0000-0001-8630-3114) to “L W Popple” in Ozymandias.

Data sources

used several different strategies to convert data into the triples required for the knowledge graph. If the source data was in the form of CSV files (e.g., the Australian Faunal Directory) it was imported into a MySQL database, and PHP scripts were written to further clean the data and map it to any external identifiers. Once the data was cleaned and linked, a PHP script was used to export the data in N-triples format.

Several sources of data (Atlas of Living Australia, CrossRef, ORCID, Wikispecies, and Biodiversity Literature Repository) were accessed via their APIs. For ALA a list of all animal taxa was obtained from the ALA web site, then the JSON record for each taxon was harvested. For CrossRef, data was harvested for just those DOIs found by the bibliographic string to DOI mapping process described above. These DOIs were also submitted to a custom script that queried the ORCID database to discover whether any authors had works with those DOIs in their ORCID profile. If this was the case, the corresponding ORCID profile was downloaded. Each DOI was also used as a query term for searching Wikispecies using its API with the “list” parameter set to “exturlusage” to find wiki pages that mentioned that DOI. Pages found were retrieved in XML format using the API, any references on that page parsed and converted into JSON. All JSON documents obtained from these sources were stored in CouchDB databases (http://couchdb.apache.org) and custom CouchDB views were written in Javascript to convert the JSON documents into N-triples.

By default Ozymandias treats individual publications as a single, monolithic entity. However, some publishers such as PLOS and Pensoft provide DOIs for component parts of an article, such as individual figures. Egloff et al. (2017) have argued that even if a taxonomic article itself is copyrighted, the individual figures are not eligible for copyright, and hence extract and assign DOIs to large numbers of figures extracted from journals such as Zootaxa. These figures, together with ones sourced from open access journals are available through the Biodiversity Literature Repository (http://plazi.org/resources/bibliography-of-life-bol/biodiversity-literature-repository-blr/) (BLR). The BLR is hosted by Zenodo (https://zenodo.org) and each publication and figure has a unique identifier (typically a DOI), and metadata for each publication and figure is available as JSON-LD. This means data from the BLR can be directly incorporated into a triple store. However for this project I wanted just a subset relevant to publications on the Australian fauna, and so I created a CouchDB version of the BLR and wrote scripts to match publications from the AFD to the corresponding record in the BLR. Metadata for each matching publication and its associated figures were then retrieved directly from Zenodo.

Knowledge graph

The knowledge graph was implemented as a triple store using Blazegraph 2.1.4 (https://www.blazegraph.com) running on a Windows 10 server, with a nginx web server (http://nginx.org) acting as a reverse proxy. N-triples for different categories of data (e.g., taxa, publications, etc.) were partitioned using named graphs corresponding to the data sources and uploaded to the triple store. There were a total of nine named graphs, the two largest (6,548,348 and 1,596,986 triples) correspond to the publications and taxa, respectively. The remaining graphs contained triples from the sources used to create crosslinks between publications and people. The use of named graphs made it easier to manage sets of data, for example the bibliographic data could be deleted and reloaded by simply deleting all triples in the corresponding named graph, rather than having to delete the entire knowledge graph. It also facilitated some queries, such as author matching across multiple data sources where distinguishing between data source was an essential part of the query.

Search

Being able to simply search for relevant documents by typing in one or more terms is a feature users expect from almost any web site. To implement search, basic information on taxa and publications was encoded into a simple JSON document (one per entity) and these JSON documents were indexed using an instance of Elasticsearch 6.3.1 hosted on Google’s Compute Engine.

Web interface

Designing a semantic web browser to display a richly interconnected data set is a challenging task (Quan & Karger, 2004). For Ozymandias the goal was to have a simple interface which encouraged the user to explore connections between taxa, publications, and people. Apart from the home page, there are two main page types in the web interface for Ozymandias. The first is the search interface which is a simple list of the entities that best match the search terms. Clicking on any member of that list leads to the second page type, which is a display of the entity itself. This display comprises three columns. The left column displays core facts about the entity. These are typically triples that have the entity as their subject, or are one edge away in the knowledge graph (such as thumbnail images), and so can be retrieved from the knowledge graph using either SPARQL DESCRIBE or CONSTRUCT queries. The middle column displays connections between the main entity on the page and related entities in the knowledge graph (such as authors of a paper, taxonomic names mentioned in a work, etc.), and is populated by SPARQL queries. The rightmost column is used to display the result of searching external sources for information relevant to the entity displayed on the page. Hence, unlike columns one and two, these queries are not SPARQL queries to the local knowledge graph.

Results

Ozymandias can be viewed at https://ozymandias-demo.herokuapp.com. Source code is available on GitHub (https://github.com/rdmpage/ozymandias-demo). Below I describe the web interface to Ozymandias, and outline some of the exploratory analyses that can be undertaken using the underlying knowledge graph. Where the results are based on SPARQL queries, those queries are listed in the Supplementary material.

Web interface

A screenshot of the web interface is shown in Fig. 5. This shows the three-column layout used to display an entity, its relationships within the knowledge graph, and any known external relationships.

Figure 5 Web interface to Ozymandias knowledge graph displaying information for an article.

The left column displays a summary of the article, and a PDF viewer (only available if content is freely accessible). The middle column displays related content from the knowledge graph, such as taxa mentioned in the article. The right column shows the result of searches in external web sites for related information, in this case is displays the identifier for Wikidata item that corresponds to this article. To view this page live go to https://ozymandias-demo.herokuapp.com/?uri=https://biodiversity.org.au/afd/publication/3e0c1402-de05-4227-9df3-803e68300623.

The first example is a publication, in this case (Nakabo, 1982). The first column summarises basic data about the publication, and if the full text is available it is displayed using either a PDF viewer, or a simple image viewer in the case of scanned images. The second column lists taxa associated with the publication. For publications with identifiers such as DOIs the third column displays whether a record with that DOI exists in external sources such as Wikidata and ORCID.

The second example (Fig. 6) displays information for an author, including a list of publications, coauthors, journals the author publishes in, and a summary of their taxonomic expertise. This later diagram is computed by using a SPARQL query to find the top 20 taxa the author has published on. For each taxon the query uses a property path expression to retrieve the list of higher taxa each taxon belongs to, and a Javascript script assembles those lists into a tree. The third panel displays the results of matching the author to author identifiers using external web services, in this case discovering the author’s ORCID id and entry in Wikidata.

Figure 6 Information about an author displayed in Ozymandias.

The left column lists the author’s publications, the middle column uses the knowledge graph to identify coauthors, venues for publication, and the taxonomic expertise of the author, the right column displays information from external sources. To view live go to https://ozymandias-demo.herokuapp.com/?uri=https://biodiversity.org.au/afd/publication/%23creator/l-w-popple.

Figure 7 shows the view of a taxon, in this case genus Acupalpa Kröber, 1912. We see the member species of the genus, the taxonomic hierarchy of the genus (generated using a SPARQL property path query) and, where available, a thumbnail image from the ALA. The second column lists the taxonomic names associated with the genus, together with the publications that made those names available. The third column shows the results of mapping the taxon to one or more external taxonomic databases, in this case GBIF.

Figure 7 Information about the genus Acupalpa Kröber, 1912 displayed in Ozymandias.

The display includes the species in the genus, details about the publication of the name Acupalpa, and a link to the taxon in GBIF. Live version at https://ozymandias-demo.herokuapp.com/?uri=https://bie.ala.org.au/species/urn:lsid:biodiversity.org.au:afd.taxon:111fc7e9-0265-453e-8e60-1761e42efc9a.

Wherever possible, Ozymandias uses thumbnail images from ALA to illustrate taxa. However, many taxa lack images. Figure 8 shows an example where the ALA has no image for a taxon (Trigonopterus cooktownensis). Because the taxon, its name, the publication, and the figures in that publication extracted by the Biodiversity Literature Repository are all part of the knowledge graph, we can traverse the graph and discover that an image for that species was published in Riedel & Tänzler (2016).

Figure 8 Augmenting data using knowledge graph.

The Atlas of Living Australia did not have an image for Trigonopterus cooktownensis at the time it was harvested by Ozymandias, hence the “?” displayed in the square in the left column. However, the original description of that species did include images which are available in the Biodiversity Literature Repository, and hence are displayed by Ozymandias in the middle column. Live example https://ozymandias-demo.herokuapp.com/?uri=https://bie.ala.org.au/species/urn:lsid:biodiversity.org.au:afd.taxon:14feec1f-9d2a-496b-9b98-ec691289b5ce.

Strings to things

Most of the work on data cleaning and linking was devoted to matching string representations of publications to the corresponding digital identifiers. The result of this matching provides us with an estimate of how many publications have been digitised and hence are potentially available online.

Figure 9 shows the distribution of publications over time, together with the numbers that have been matched to digital identifiers. The pattern of publication shows three prominent dips. The first two correspond to the two world wars in the twentieth century, the third dip occurs from the mid-1990’s to the present day. Given that the AFD is retrospectively collecting publication data, it is not clear to what extent this decline in recent publications represents an actual decline in activity versus a under sampling the most recent literature.

Figure 9 Plot of publications over time.

As well as the total number of publications for each year, the chart shows the numbers of publications that have a digital identifier (DOI, BioStor, or JSTOR) or have a PDF available online.

Many publications lack a digital identifier, suggesting that a considerable amount of the relevant literature has not been digitised. However, this may be overstated as the matching was done by a single individual working to a deadline (in this case the Ebbe Nielsen Challenge submission date). As more effort is expended on matching records the gap between the number of publications and the number of publications online is likely to decrease.

Linking authors to identifiers

We can measure the uptake of ORCID ids for researchers working on the Australian fauna by using DOIs to match works in the knowledge graph to works in the ORCID database. ORCID was launched in 2012, for period from 2012 to the present day the knowledge graph contains 2302 distinct author names. Matching DOIs for the works those authors published to the ORCID database shows that 346 (15%) of authors publishing in that time period have ORCID ids. This number is likely to be an underestimate as not all works in ORCID have DOIs (and ORCID records sometimes omit DOIs for works that have them), but it suggests limited adoption of ORCIDs amongst taxonomists and other biodiversity researchers.

Changes in taxonomic publications over time

To explore the publication history of taxonomic research on Australian animals for each decade from 1820 to 2020 I found the ten journals that had the most articles in the knowledge graph. The numbers of articles in each journal were plotted for each decade (Fig. 10). Over time different journals have been dominant venues for publishing taxonomic work. In the 18th century British or other European journals dominated, such as Proceedings of the Zoological Society and Annals and Magazine of Natural History, although the local journal Proceedings of the Linnean Society of New South Wales (establish 1875) was a major outlet for taxonomic work. In the mid to late 20th century Australian journals, typically published by museums or by the Commonwealth Scientific and Industrial Research Organisation (CSIRO) were the primary venues for taxonomic papers on the Australian fauna. However, the last decade has seen the spectacular rise of the “megajournal” Zootaxa, published in New Zealand but with global coverage. Hence, taxonomic publication in Australia has gone from an early colonial period where much of it was published overseas, to a national period where many papers were published in local journals, culminating in the present situation where international journals such as Zootaxa and, to a lesser extent Zookeys, dominate.

Figure 10 Patterns of publication of taxonomic work on Australian animals 1820–2020.

For each journal sparklines depict the numbers of publications for each. Non-Australian journals are highlighted with a grey background. The 19th and early 20th centuries are dominated by European journals, by the mid 20th century most taxonomy was published in Australian journals, more recently international journals such as Zootaxa are increasingly important.

Citations and taxonomy as long data

Taxonomy is a “long data” discipline (Page, 2016b). In some scientific fields published papers have a short citation half-life and hence are relatively ephemeral, quickly losing their relevance as the “research front” moves on (De Solla Price, 1965). The rise of academic search engines such as Google Scholar may increase the discoverability of the older literature (and hence increasing its likelihood of being cited (Verstak et al., 2014), but for many fields the older literature fades from importance. In contrast, the taxonomic literature is essentially ageless—any published work is potentially relevant. Part of this relevance reflects the importance of priority in biological nomenclature, given competing names for the same taxon in general the oldest name wins. Another factor is the sheer number of species and the relative paucity of published knowledge on many of those species. May (1988) estimated that for publications in the period 1978 to 1987 for insects there were on average 0.02 papers per species per year, for beetles it was 0.01 papers. Hence a researcher may have to search back through a hundred years of literature in order to find mention of a specific beetle species.

To explore the citation graph for publications on the Australian fauna I queried each citation relationship for the dates of publication of the citing and the cited works. The relationship between these two dates (Fig. 11) highlights the enduring value of the older taxonomic literature. If taxonomic work cited only recent publications then the points in Fig. 11 would fall on or close to the diagonal. However, even papers published recently (top right of the chart) cite older literature (represented by the vertical columns of dots below each year), and hence much of the area below the diagonal is occupied.

Figure 11 Dates of publication of works cited against the date of publication of the cited work.

Each point represents the (x, y) pair (publication date, cited publication date). All cited works must, by definition, be published in the same year or earlier, and hence the points fall on or below the diagonal. The few points that are above the diagonal represent errors in CrossRef’s metadata.

History of species discovery in different taxonomic groups

The knowledge graph enables exploration of the taxonomic history of any taxon of interest. Pullen, Jennings & Oberprieler (2014) recently reviewed the history of weevil taxonomy in Australia. Ozymandias has some 3,958 accepted weevil species. For each accepted taxon in the ALA classification I used a SPARQL query to retrieve the date the species was originally described, and the dates where then grouped by year. The plot of cumulative numbers of accepted species over time (Fig. 12) closely matches that reported by Pullen et al.

Figure 12 Plot of the history of species discovery for Australian weevils.

The solid line is the cumulative number of weevil species that are currently accepted. The vertical bars record the number of new weevil species names published each year. Note the relatively modest increase in names and taxa since the 1930’s.

The same chart also shows the number of weevil species names published each year, including synonyms. This chart shows that the bulk of weevil discovery took place in the mid-19th to mid-20th centuries. The sharp drop in species discovery since the 1930’s may indicate that the bulk of the Australian weevil fauna has been described, but this seems unlikely given that weevils are typically small and cryptic, and many species in leaf-litter and other habitats may remain undiscovered (Stork et al., 2008; Riedel & Tänzler, 2016).

These same queries can be used on other taxonomic groups, enabling us to compare the state of knowledge for different taxa. For example, the land snail family Camaenidae (Fig. 13) shows a similar pattern of discovery in the mid-19th to mid-20th centuries to that seen in weevils. However, in contrast to weevils these snails have been the subject of ongoing study with over 200 new species being described in the last decade (Köhler, 2010; Köhler, 2011), a rate of discovery that shows no sign of declining.

Figure 13 Plot of the history of species discovery for Australian snails in the family Camaenidae.

The solid line is the cumulative number of camaenid species that are currently accepted. The vertical bars record the number of new camaenid species names published each year. In contrast to the weevils (Fig. weevils) new Camaenidae species are continuing to be discovered.

Discussion

Building a knowledge graph requires mapping textual representations of entities to identifiers that are shared across data sources (“strings to things”). This mapping is tedious and time consuming to construct, and in a time limited project such as a challenge entry like Ozymandias the mapping is likely to be incomplete before the deadline for the project. Despite its necessarily incomplete state I think the project illustrates some of the ways a network approach can enrich our knowledge of a topic. The web interface exposes many more connections between taxa, publications and people than are evident in the Atlas of Living Australia and Australian Faunal Directory that were used as source databases.

The underlying knowledge graph can be used to support queries exploring the history of taxonomic publishing and discovery. Some of these queries could be used to help prioritise future work. For example, the pattern of citations (Fig. 11) confirms that the older taxonomic literature is still relevant today, reinforcing the case for digitising the legacy taxonomic literature. We could further explore the citation data to prioritise which journals should be scanned first: for example, by focusing on those journals that have been cited the most. Given that the bulk of taxonomic publications in the 20th century appeared in Australian journals, initiatives such as the Biodiversity Heritage Library in Australia would seem well placed to make the case that this work should be scanned and made openly available. Citation counts can also be used more directly. For example, the International Institute for Species Exploration annually issues a manually curated list of the “top 10” species discovered the previous year (https://www.esf.edu/top10/). Such a list could be automatically generated from a knowledge graph using, for example, the number of citations (or other measures of attention) that each work publishing a new species has received.

Some analyses of the knowledge graph are more focussed on the state of the knowledge graph itself. For example, querying for author identifiers such as ORCIDs reveals a limited uptake of that identifier. This has implications for proposals to use ORCID as the basis for tracking the broader activities of taxonomists, such as specimen collection and identification. Perhaps the development of tools such as David Shorthouse’s Bloodhound (https://bloodhound-tracker.net) may help raise awareness of the possible benefits of authors registering with ORCID.

Expanding the knowledge graph

The knowledge graph in Ozymandias features only a subset of the entities depicted in earlier work sketching the “biodiversity knowledge graph” (Page, 2013; Page, 2016a). There are several entities that are obvious candidates to be added to Ozymandias, such as specimens and nucleotide sequences. However, the number of specimens that could potentially be added has implications for the scalability of the knowledge graph. Bearing this in mind, we could add a subset of specimens, such as type specimens, or those which have been sequenced. Fontaine, Perrard & Bouchet (2012) reported that the average lag time between the discovery of a specimen representing a new species and the description of that species is 21 years. The generality of this observation could be evaluated using a knowledge graph that contains both the taxonomic literature and type specimens with dates of collection.

The Biodiversity Literature Repository highlights the potential of treating scientific articles not as monolithic entities but rather as assemblages of component parts, including figures. We can drill down further and start to annotate individual parts including fragments of text. The idea of annotating and interlinking fragments of text has a long history, pioneered by people such as Ted Nelson (Dechow & Struppa, 2015), and tools such as Hypotheses.is (https://web.hypothes.is) now make this possible. We could view the “micro citations” used by taxonomists to specify the page location of a taxonomic name as a form of annotation, hence a logical next step is to map these micro citations onto publications in the knowledge graph so that we can locate these micro citations in the context of the taxonomic literature that they refer to.

The future of knowledge graphs

To the extent that Ozymandias is judged to be a success it suggests that knowledge graphs have potential to improve the way we aggregate and interface with biodiversity data. However, it is worth noting that the biodiversity informatics community has been aware of knowledge graphs and semantic web technologies for a decade or more, and several taxonomic databases have been serving data in RDF since the mid-2000’s. Yet it is hard to point to successful applications of these approaches to the study of biodiversity, and there has been limited uptake of linked data beyond a few databases. It is also not clear which is more valuable: (a) the insights that could be obtained from querying the graph, or (b) the cleaning and reconciliation of the data takes place before the graph can be constructed. Despite this, I think there are some areas where a biodiversity knowledge graph could potentially be very useful.

There is growing concern within biology in general (McDade et al., 2011) and in taxonomy in particular, that existing measures of the output of researchers, such as citations, are poor metrics of activity (Tancoigne & Ollivier, 2017). There are also concerns that existing data aggregators do not pay enough attention to tracking the provenance and authorship of information (Franz & Sterner, 2018). Researchers may do much more than write papers, they may clean, prepare, and publish datasets, collect specimens, curate collections, identify specimens, etc. Keeping track of these activities is greatly facilitated by the use of stable identifiers for people and the objects they work with (e.g., specimens, collections, datasets), and a knowledge graph would be an ideal data structure to quantify the work done, and trace the provenance of data and associated annotations. Hence, it may be that the best way to bootstrap the adoption of biodiversity knowledge graphs is to focus on the implications for being able to give appropriate credit to researchers for all the activities that they undertake.

There is considerable enthusiasm for the potential of identifiers to help evaluate research (Haak, Meadows & Brown, 2018) and yield insights into the behaviour of researchers (Bohannon, 2017). However, the ease with which measures of research activity (such as citation-based measures) switch from being tools for insight into targets to be met suggests we should consider the possibility that metrics developed to create incentives to build knowledge graphs may ultimately harm the researchers being measured.

Beyond internal drivers, such as documenting the provenance of taxonomic information, and quantifying the contributions of researchers, there are also external drivers for considering knowledge graphs. Wikidata (https://www.wikidata.org) (Vrandečić & Krötzsch, 2014) is an open, global knowledge graph with an enthusiastic community of editors, and many of the entities taxonomists care about are already included in the graph, such as taxa, people, and publications. Projects such as Scholia (https://tools.wmflabs.org/scholia/) (Nielsen, Mietchen & Willighagen, 2017) have already demonstrated the potential of Wikidata to explore the output of scholars. Another potential use of Wikidata is to help define the scope of a knowledge graph. Anyone constructing a knowledge graph rapidly runs into the problem of scope, in other words, when do you stop adding entities? Once we move beyond specialist knowledge in a given field (such as specimens, rules of nomenclature, sequences and phylogenies) and include more generic entities that other communities may also be interested in (such as publications, natural history collections, people) we reach the point at which we can stop constructing our graph and defer to Wikidata. Hence a key part of the future development of biodiversity knowledge graphs will be to determine the extent to which Wikidata and its community can be responsible for managing biodiversity-related data.

Conclusions

Ozymandias is a biodiversity knowledge graph that contains information on Australian animal taxa and their names, taxonomic publications, the authors of those publications together with their interrelationships, such as publication, citation, and authorship. The knowledge graph is implemented as a triple store, and contains over 9 million triples. A web interface to this triple store makes it possible navigate through the data from various perspectives, including taxonomy, publications, or authors. This interface was submitted as an entry in the 2018 GBIF Ebbe Nielsen Challenge.

Supplemental Information

Supplemental Information 1 SPARQL queries used to generate Figs. 9–13

Click here for additional data file.

Data S1 Data for Fig. 9

For each year the table lists the total number of publications in that year, and how many had digital identifiers (DOI, BioStor, JSTOR) and how many are available as a PDF file.

Click here for additional data file.

Data S2 Data for Fig. 10

Decadal counts of publications in selected journals.

Click here for additional data file.

Data S3 Data for Fig. 11

Each row records the date of publication of an article (”from”) and the date of publication for a work cited by that article (”to”).

Click here for additional data file.

Data S4 Data for Fig. 12

Numbers of described species, and cumulative total number of accepted species of weevils per year.

Click here for additional data file.

Data S5 Data for Fig. 13

Numbers of described species, and cumulative total number of accepted species of land snails (family Camaenidae) per year.

Click here for additional data file.

Constructing the knowledge graph described here would have been impossible without the wealth of freely available and open source software used in the project. Furthermore, it should be obvious that none of this would have been possible without the centuries of taxonomic research by generations of researchers, and the recent efforts to make that research digitally accessible via projects such as Atlas of Living Australia and the Australian Faunal Directory. I’m also indebted to GBIF for running the 2018 Ebbe Nielsen Challenge which gave me a hard deadline to work towards (and the pleasant surprise of being a joint winner). I thank Steve Baskauf and Joel Sachs for feedback on the project, and for invitations to present Ozymandias to their colleagues. Hilmar Lapp, Campbell Webb, and Anne Thessen provided very helpful and constructive reviews of the manuscript.

Glossary

AFD Australian Faunal Directory, a database of Australian animal taxa and taxonomic publications

ALA Atlas of Living Australia, a database of Australian biodiversity

BHL Biodiversity Heritage Library, a database of scanned biodiversity literature

BLR Biodiversity Literature Repository, a database of biodiversity literature and images

CSV Comma separated values, a file format for tabular data

DOI Digital Object Identifier, a globally unique identifier for digital objects

GBIF Global Biodiversity Information Facility

HTTP Hypertext Transfer Protocol, the underlying protocol of the web

HTTPS A secure version of HTTP

ISSN International Standard Serial Number, used to uniquely identify periodicals

JSON Javascript Object Notation, a format for representing data in Javascript

JSON-LD JSON linked data, a lightweight format for linked data

Linked data A method of publishing structured data as RDF on the web

LSID Life Sciences Identifier, a globally unique identifier for biological entities

Named graph A subset of a triples in a triple store

ORCID A globally unique identifier for researchers

RDF Resource Description Format, a standard model for data interchange on the web

SPARQL A query language for RDF

SICI Serial Item and Contribution Identifier, a globally unique identifier for a journal article

Triple A data item comprising a subject, predicate, and object

Triplestore A database that stores RDF triples and can be queried using SPARQL

XML Extended markup language

UUID Universally unique identifier, a 128 bit globally unique identifier

Additional Information and Declarations

Competing Interests

Author Contributions

Data Availability

The authors declare there are no competing interests.

Roderic D.M. Page conceived and designed the experiments, performed the experiments, analyzed the data, contributed reagents/materials/analysis tools, prepared figures and/or tables, authored or reviewed drafts of the paper, approved the final draft.

The following information was supplied regarding data availability:

Source code for the interface is available from GitHub (https://github.com/rdmpage/ozymandias-demo). Source code for the scripts used to harvest and clean the data used to populate the knowledge graph is available from https://github.com/rdmpage/oz-afd-export, https://github.com/rdmpage/oz-ala-harvest, https://github.com/rdmpage/oz-csl, and https://github.com/rdmpage/oz-wikispecies.

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
