# Peer review of "Ozymandias: a biodiversity knowledge graph"

_PeerJ, doi:10.7717/peerj.6739_

## Round 0.1 · original submission · Minor Revisions

Nicely written manuscript. Aside from the reviewer's comments, I only have a few minor additional ones (line numbers pertain to PDF):

- L145-147 "DOIs are dereferenced (“resolved”) using the web proxy
https://doi.org. This proxy recently switched from the HTTP to the HTTPS protocol, immediately rendering out of date any DOIs stored URLs starting with the prefix “http://”." I don't think that's fair, or perhaps define what you mean by "out of date". If understood as "do not resolve any more" as readers might be inclined to, then the assertion is not true. It seems your point here is that over time there have been different official recommendations for the canonical representation of DOIs (from non-prefixed, to using http://dx.doi.org, to http://doi.org, etc), and that therefore such representations found in the literature at different times may not match up even if the DOI part alone does. Which is a fair point, so I'd suggest to clarify.

- Perhaps I've missed the respective definitions, but it seems that several acronyms (especially technical ones) are used without prior definition. Consider adding a table of acronyms that spells out each one.

- Reviewer 1 notes that the path from Publication to Person goes through the (is it the same?) "creator" predicate twice. I suggest you clarify that this is prescribed by the schema.org model (if it is?) so as to help avoid (even if only minor) reader confusion. Reviewer 1 also suggests adding prefixes (namespaces); I agree this would help with disambiguation (even if only by making it clear that there is nothing to disambiguate because the labels happen not to be ambiguous).

·

Basic reporting

This paper describes a project of the author’s to reconcile data from a number of online data sources, link the data together using RDF triples, and build a web interface to browse the data. The data are: taxonomic names of the fauna in the Atlas of Living Australia, and articles and their authors that published these names. The article also presents some pertinent analyses of the taxon/article/author data once they had been assembled. The project itself is a valuable contribution in the field of biodiversity informatics and taxonomy, and the paper describes the project clearly and comprehensively.

My criticisms are relatively minor. Some relate to the presentation (here), some to engineering choices (section 2, below), some to conclusions (section 3, below).

1. The Abstract does not reflect the full substance of the paper. Some mention of each of the elements of “reconciliation, augmentation and meta-analysis” should be added. The analyses that the author performed on Ozymandias data are themselves small but significant contributions to taxonomic science and should be mentioned in the abstract.

2. The paper shows signs of having been submitted perhaps before one final editorial check through! There are more typos than there should be, and several missing citations.

3. The formatting of in-text citation is often incorrect. I see that PeerJ does not enforce a particular citation style, but I imagine the in-text citations should be performed by the author. E.g., “published in (Riedel & Tänzler, 2016)” should be “published in Riedel & Tänzler (2016)”, etc.

4. The paper seems just a touch too long (10-15%). Some detail could be omitted: for example, some of the the details of DOI dereferencing (LL 145-152), some of the details of the web interface (LL 296-326), and some of the text from LL 163-170 and LL 225-237. On the other hand, the details of reconciling author names (LL 204-216) is valuable and probably represents a lot of the time that went in the project.

5. A trivial question, but one most readers will have: Why the name Ozymandias? After the poem by Shelley? A short answer to that question would be appropriate in the paper.

6. There are few mentions of other biodiversity knowledge graphs, suggesting that Ozymandias is one of the very first. If we include any sharing of biodiversity data as LOD, then there are many (e.g., OpenBioDiv, Minami et al (2011, “Towards a Data Hub for Biodiversity with LOD”), Mai et al. (2011, “Linked Open Data of Ecology (LODE): A New Approach for Ecological Data Sharing”), http://bioimages.vanderbilt.edu/, taxonconcept.org, etc).


Line by line:

L 38. “representation of a knowledge graph” -> “implementation of the basic unit of a knowledge graph”

L 40. maybe list a few other ways to model a knowledge graph (KG)?

L 49. Typo: “identify” -> “identity”

L 79-80. It seems to me that reconciliation is not “built on top” of a KG, but is a key prior step in creating that KG.

L 117. Typo: “adopted to because” -> “adopted because”

L 131. “(Vicki Tardif Holland & Jason Johnson, 2014)” -> “(Holland & Johnson, 2014)” and in references list.

L 250 Typo: “write” -> “wrote”

L 378. Check commas, parentheses around the citation.

L 445. Missing citation.

L 453. Missing citation.

L 479. Missing citation.

LL 481-490, and LL 492-497. Good points!


Figures:

Fig 1. Would be helpful to add the namespace to the classes and properties. Also the layout of the figure could be improved to save space and make it a bit more “artistic”.

Fig 3. Could be omitted or made supplemental information.

Fig 4. Could be omitted or made supplemental information. Well described in text.

Figs 5, 6, 7, 8. Could be omitted or made supplemental information. While a great deal of work went into the design of the web interface, these views are available on the web.

Fig 9. Not sure about the PeerJ color figure policy, but this figure could easily be made into a black-and-white figure.

Fig 10. Legend typo: “publications for each.” -> “publications.”? While “sparklines” (I had to look this term up) don’t have axes, this figure would better served by adding axes: “decade”, vs. “number of publications per decade”.

Fig. 11. Good figure!

Figs 12 and 13. It would be nice to also see the number of (current) synonyms as well (using both a cumulative line and as bars). This would require a new query and new figure and is not vital (for me), but would enhance the graphs. Some comment/discussion of this would also be needed in the text.


In summary, the article:

* Is clearly written.
* Gives sufficient context with citations (but see comments above).
* Is acceptably structured (it is hard sometimes to fit description of software into a standard scientific article format, but the author has done this well).
* Contains only figures relevant to its content, though some may not be needed (see above).
* Is self-contained and an appropriate unit.

Experimental design

Since this is a software product rather than experiment, I’ll read “engineering design” for “experimental design”. Overall, Ozymandias appears to me to be very well-designed, and the choices made are nicely laid out in the text of this paper (if only all software/data products had an associated paper with this level of documentation of choices!).

I’m not a big fan of the way Schema.org models roles (http://blog.schema.org/2014/06/introducing-role.html), with it’s duplication of the `creator` property, and so would probably have tried to find another ontology/schema to model article authorship (Bibo?). However, I accept that modeling ordered lists in graphs is never easy, and also accept the value of choosing widely-used schemas (like Schema.org) where possible. One compromise if one needs to use a more domain-specific vocab is to specify the Schema.org classes in addition to the classes of the other vocabulary, which aids in discovery if not data integration.

The associated analyses (Figs. 9-13) were valuable and well described. I especially enjoyed the discussion of taxonomy as “long data”. It would have been nice to see a few more comparisons of the results of these analyses with those done by other authors (the comparison with Pullen et al. 2014 excepted).

The comparison of weevils vs. snails (Fig 12 vs. 13) was interesting and well described (LL 404-409).

Validity of the findings

The conclusions of the paper are valid, and all the data necessary to fully reproduce the software product, and the analyses, are provided (via Github, the included SPARQL queries and outputs, and via the ability to run _any query_ via the SPARQL endpoint).

A couple of comments that the author may chose to further reflect on in the text:

1. In a couple places the author states that Ozymandias was made to a deadline and so the work of matching data from different sources had to be halted. This reconciliation step is what makes the construction of the KG possible, and is also probably the most time consuming. Even though it was not needed for this product, the author alludes to the process (and problem) of matching taxonomic names (LL 182-184), and this is a critical and time-consuming step in the construction of most biodiversity knowledge graphs. Can this reconciliation phase, which requires many “bespoke” solutions to be engineered, really be scaled up to, e.g., GBIF-scale?

2. Without calling into question the value of Ozymandias and this paper, I wonder if the additive value of Ozymandias is really a result of it being based on a graph data model (cf. LL 422-423), or of the data reconciliation steps that it required. The author’s other major data/software contributions in biodiversity informatics (e.g. iSpecies, BioNames, BioStor) are not built on graph databases (as far as I know) and have similar user interfaces. To truly demonstrate the value of the KG nature of Ozymandias, I think one would have to show that a) users were accessing the end point to ask queries not asked by the web interface, and b) that these queries feed directly into graph-like analyses and products (i.e., they need the RDF output of Ozymandias, and could not have been run as SQL queries, should there have been a SQL Query interface to the underlying Ozymandias data). Another way to demonstrate the “graph power” of Ozymandias would have been to provide some original metadata served as RDF, via “live” content negotiation with triples having an Ozymandias URI. Finally, perhaps the graph data could be contributed to a 3rd party triplestore (Wikidata, https://datahub.ckan.io/) which would at least provide the opportunity for users to integrate Ozymandias’ new data with other graph data.

The author alludes to the question of how valuable graph data really is for biodiversity in LL 468-474, and the jury is out on where biodiversity LOD will go. But perhaps it might be more explicitly recognized in the paper that _currently_ it is really the data integration across data silo that liberates data rather than the KG nature of the output.

·

Basic reporting

The author meets all basic reporting requirements.

Experimental design

The author meets all experimental design requirements.

Validity of the findings

The author meets all validity requirements.

Additional comments

I enjoyed this paper and I think it can be published with some VERY minor changes.
1. Sometimes it is important to find the last author of a paper. How would one do that using the numerical roleName?
2. This probably isn't a problem, but it seems weird to me that a TaxonName would be a synonym of a TaxonConcept. Wouldn't another TaxonConcept be the synonym and the synonymous TaxonName attached to that?

Small typos:
Line 16: should be "and examples of its application"
Lines 55-57 and 71: I don't understand why CrossRef, Altmetric, ORCID, and Schema.org are in parentheses and quotes
Line 90: I suppose its the author's preference, but I think of "data" as plural
Lines 156-157: should be "themselves have DOIs. Hence, if we"
Line 174: should be "cases the sources of data"
Lines 187-188: subordinate clauses at the beginning of a sentence should be followed by a comma. Example "For the works in AFD, I searched". There are many of these throughout the paper.
Line 241: should be "Egloff et al. (2017) have"
Line 257: Can you mentioned how many named graphs you had? Their sizes? Just a point of curiosity
Line 391: should be "Figure 11 would"
Line 398: should be "Pullen, Jennings & Oberprieler (2014) recently"
Line 408: should this be "species in leaf-litter and other habitats"?
Line 445: Is the reference to Shorthouse a pers. comm.?
Line 479: Did you forget to replace "cite citation papers" with actual citations?
Line 525: I didn't see anything but a readme in the oz-wikispecies repo. Is this correct?
Line 677: should be "in this case it displays"
Line 694; should this be "data using the knowledge"?
Line 726: should be "Fig. 12"

---

## Round 0.2 · accepted · Accept

One note after re-checking the figures, in Fig. 4 the "value" relationship is directed from what appears to be the value to the "propertyValue" node (which is counterintuitive), whereas in Fig. 3 it seems to be directed in the opposite (and intuitive) direction. Are both correct? This can be resolved in production if needed.

#